# Effect of a Digitally-Enabled, Preventive Health Program on Blood Pressure in an Adult, Dutch General Population Cohort: An Observational Pilot Study

**DOI:** 10.3390/ijerph19074171

**Published:** 2022-03-31

**Authors:** José Castela Forte, Pytrik Folkertsma, Rahul Gannamani, Sridhar Kumaraswamy, Sipko van Dam, Jan Hoogsteen

**Affiliations:** 1Department of Clinical Pharmacy and Pharmacology, University Medical Center Groningen, University of Groningen, 9711 LM Groningen, The Netherlands; 2Ancora Health B.V., 9711 LM Groningen, The Netherlands; pytrik@ancora.health (P.F.); rahul@ancora.health (R.G.); sridhar@ancora.health (S.K.); sipko@ancora.health (S.v.D.); janhoogsteen@ancora.health (J.H.); 3Department of Endocrinology, University Medical Center Groningen, University of Groningen, 9711 LM Groningen, The Netherlands; 4Department of Neurology, University Medical Center Groningen, University of Groningen, 9711 LM Groningen, The Netherlands

**Keywords:** hypertension, blood pressure, lifestyle, digital health, ehealth, prevention, behavioral change

## Abstract

Worldwide, it is estimated that at least one in four adults suffers from hypertension, and this number is expected to increase as populations grow and age. Blood pressure (BP) possesses substantial heritability, but is also heavily modulated by lifestyle factors. As such, digital, lifestyle-based interventions are a promising alternative to standard care for hypertension prevention and management. In this study, we assessed the prevalence of elevated and high BP in a Dutch general population cohort undergoing a health screening, and observed the effects of a subsequent self-initiated, digitally-enabled lifestyle program on BP regulation. Baseline data were available for 348 participants, of which 56 had partaken in a BP-focused lifestyle program and got remeasured 10 months after the intervention. Participants with elevated SBP and DBP at baseline showed a mean decrease of 7.2 mmHg and 5.4 mmHg, respectively. Additionally, 70% and 72.5% of participants showed an improvement in systolic and diastolic BP at remeasurement. These improvements in BP are superior to those seen in other recent studies. The long-term sustainability and the efficacy of this and similar digital lifestyle interventions will need to be established in additional, larger studies.

## 1. Introduction

Hypertension is one of the most important and prevalent risk factors for cardiovascular and kidney disease development, with studies showing that up to 22% of myocardial infarctions (MI) in Europe are related to hypertension, with hypertensive individuals having almost double the risk of MI compared to those with no history of hypertension [1,2,3]. Worldwide, it is estimated that at least one in four adults suffers from hypertension, with this number being expected to increase as populations grow and age [4].

With regards to etiology, while high blood pressure (BP) possesses a heritable component, lifestyle risk factors contribute significantly to hypertension and can have a substantial effect beyond genetic predisposition [5]. Recently, a compound healthy lifestyle score was strongly inversely associated with both systolic blood pressure (SBP) and diastolic blood pressure (DBP), irrespective of the underlying genetic risk: participants with a favourable lifestyle had 4 to 5 mmHg lower systolic BP across of strata of genetic risk compared to those with an unfavourable lifestyle [6]. Older data from previous large studies of individuals from Finland, Italy, The Netherlands, UK and the US further indicate that dietary and lifestyle interventions can affect BP by as much as 20 mmHg [7]. Lifestyle modifications proven to effectively lower BP include weight loss, reduced sodium intake, increased physical activity, limited alcohol consumption, and following the Dietary Approaches to Stop Hypertension (DASH) dietary pattern, which emphasizes consumption of whole foods, in particular fruits, vegetables, low-fat dairy products, whole grains, but also poultry, fish, and nuts, accompanied by a reduction in (saturated) fats, red meat, sweets, and sugar-containing beverages [8,9].

However, while the dietary and lifestyle changes that could contribute to the prevention of hypertension are simple and well-known to the general public and are included in most national and international guidelines, there’s a lack of truly effective strategies promoting risk reduction through these lifestyle factors [10,11,12]. On the one hand, this is exacerbated by the difficulties primary care providers face in implementing advice and referral structures for lifestyle promotion. And, on the other, by the barriers faced by individuals to successfully change and maintain favourable health behaviors [13,14]. Reasons for the latter vary greatly, from limiting social constructs such as work hours, family duties, and socioeconomic factors, to personal factors such as low self-efficacy, motivation, or lack of perceived benefit [15]. A growing number of digital, app-based programs are becoming available that can support individuals in addressing these challenges [16]. Especially in the wake of the COVID-19 pandemic, the adoption of digital health technologies has progressed rapidly, with patients seeming increasingly receptive to alternatives to standard care and more willing to take greater responsibility for their health [17,18]. However, the majority of digital interventions currently available for hypertension are intended for patient self-monitoring only, which several trials have shown to have only a small effect (if any) on improving BP levels [19,20]. Applications targeted at supporting the implementation of lifestyle changes in hypertensive patients, through combined digital and human coaching, therefore remain scarce, but appear to be more effective compared to patient self-monitoring via digital applications on their own [19,20,21].

Given their potential to make prevention and management strategies outside of traditional care more effective, further research to establish the feasibility and real-world impact of these digital strategies is needed. As such, we set up this study with a two-fold goal: first, to assess the prevalence of elevated and high BP in a Dutch general population cohort undergoing a health screening, and, second, to observe the effect on BP of a subsequent self-initiated, digitally-enabled lifestyle program.

## 2. Materials and Methods

Individuals from a general population in The Netherlands who had undergone a digital, lifestyle program at Ancora Health were considered for enrolment, which was approved through a waiver issued by the University Medical Centre Groningen Medical Ethical Committee (METC#2021/488). All participants provided written informed consent to participate in the study. An overview of the study flow is provided in Figure 1.

### 2.1. Data Collection and Polygenic Risk Scoring

Participants attended a health center for a baseline assessment, where they had their systolic blood pressure (SBP) and diastolic blood pressure (DBP) measured using the InBody BIOBP750 blood pressure cuff (InBody Japan Inc., Tokyo, Japan). Values were reported to the nearest mmHg and reported as an average recording from two consecutive measurements. Patients were subsequently stratified as normotensive, or as having stage 1 (DBP between 80 and 90, or SBP between 130 and 140) or stage 2 hypertension (DBP ≥ 90 or SBP ≥ 140), according to the ACC/AHA guidelines [22]. Polygenic risk scores (PRS) were calculated using an additive model, as described in more detail in a previous publication [23]. In short, individuals were binned into deciles based on their PRS and the average disease incidence was calculated for each decile. The 1000 Genomes dataset was used as a reference panel for the linkage disequilibrium (LD) calculations [24] The LDpred tool was used to correct for LD, and minimize the risk inflation of the estimated risk through the repeated addition of the same effect across different SNPs [25]. Summary statistics files from a large GWAS conducted in another cohort was used to calculate the PRS [26]. In total, 400,016 SNPs were included to compute the hypertension PRS. Participants were then binned into deciles based on their PRS, and assigned “not elevated”, “elevated”, or “high” risk. Sex, age, self-reported presence of hypertension and medication were recorded as part of a health and lifestyle questionnaire.

### 2.2. Risk Stratification and Lifestyle Intervention

Based on a risk stratification using these data, participants followed a 16-week lifestyle coaching program rooted in the Fogg Behaviour Model, focused on nutrition, physical activity, and other health behaviors. Coaching was primarily digital, and consisted of 1-on-1 chat-based coaching, complemented by a web application and weekly progress reports with feedback. Through this approach, participants were provided peer-support and positive feedback, coached on how to acquire and maintain healthy habits and taught how to overcome barriers encountered during behavior change, and they were provided with tips and tricks on how to implement new behaviors into daily practice. Participants with abnormal baseline blood pressure received specific coaching and advice for this, including increasing fruit and vegetable consumption while decreasing the intake of red meat, alcohol and salt; gradually increasing the amount of aerobic activities performed in a week; as well as practicing stress management techniques such as yoga, meditation, and breathing exercises where required [8]. After completing the intervention, participants were given the possibility to once again come to the health centre and remeasure the health parameters they had been working on. Blood pressure values at remeasurement were measured using to the same methodology described for the initial BP measurement.

### 2.3. Statistical Analysis

Descriptive statistics were calculated to characterize the population at baseline, in terms of demographics, genetics, and blood pressure. Paired remeasurement versus baseline changes in SBP and DBP in patients who were remeasured after an average of 10 months were assessed using the Wilcoxon Signed-Rank test due to the small sample size. We also computed the percentage of participants by category of change in blood pressure from baseline to after the intervention period, from clinical threshold values to normal values, differences which were assessed using McNemar’s test. All categorical variables were reported as percentages and continuous variables as mean and standard deviation (SD). For differences in categorical variables at the cohort level, the chi-square test was used, in addition to analysis of variance tests for continuous variables. We considered a *p*-value < 0.05 as statistically significant for differences in BP and baseline characteristics. To assess whether baseline genetic risk, age, gender, and other possibly relevant lifestyle factors were predictors of changes in BP, a multiple linear regression model was tested. These included nutrition, physical activity, and stress management scores which were calculated based on eight questions for each domain, and were included as continuous variables. Sex, high genetic risk, and weight loss were modeled as binary. All data analyses were performed using R software v4.0.3 (R Foundation for Statistical Computing, Vienna, Austria).

## 3. Results

### 3.1. Prevalence of High Blood Pressure and Genetic Risk

Baseline data were available for 348 participants (Table 1). The mean baseline age was 44.6 years (sd 11.1), and 56% of the participants were women. The baseline BP for the entire cohort was 131 mmHg (sd 16.4) systolic, and 81 mmHg (sd 11.2) diastolic. Two-hundred nine participants (60%) had stage 1 hypertension or higher: 98 (28.1%) having stage 1, and 111 (31.9%) stage 2. Of these, 74 (21.3% of the total population) had a DBP between 90 and 120 mmHg, and 82 (23.6%) had a SBP between 140 and 180 mmHg. Elevated or high genetic risk was identified in 78 participants (22.4%). However, there was no significant difference in SBP (131 vs 130.6 mmHg, *p* = 0.85) or DBP (81.4, 10.33 vs 80.5 mmHg, *p* = 0.52) compared to those not at elevated risk.

### 3.2. Effect of the Lifestyle Intervention on Blood Pressure

One-hundred participants underwent a remeasurement after intervention, of which 56 had partaken in the BP-focused lifestyle program. Participants in the blood pressure-focused intervention group were on average 1.4 years older (*p =* 0.02), and had higher SBP (+6 mmHg) and DBP (+6.4 mmHg) (*p* < 0.0001). There were no significant differences in genetic risk, and only minor differences in anthropometrics, including a lower BMI in the intervention group (−0.6 kg/m^2^). No participants in the intervention group had self-reported hypertension or used antihypertensive medication.

After intervention, those with elevated SBP and DBP at baseline showed a mean decrease of 7.2 mmHg and 5.4 mmHg, respectively (*p* = 0.01 and *p* = 0.008) (Table 2).

Thirty-seven participants of the 51 (72.5%) with elevated DBP at baseline showed an improvement at remeasurement (*p* < 0.0001, Figure 2). Of these, 25 returned to within normal range, and 32 showed a decrease ≥4 mmHg (*p* < 0.0001, Figure 2). Of the 40 with elevated SBP, 28 (70%) showed improvement at remeasurement, with 13 returning to normal range (*p* < 0.001, Figure 2) and 24 showed a ≥5 mmHg decrease (*p* < 0.0001).

Neither higher genetic risk (unadjusted OR 1.19, 95% CI 0.51 to 2.86, *p* = 0.69), weight loss (OR 4.28, 0.79 to 23.24, *p* = 0.09), nor female sex (OR 0.69, 0.39 to 1.22, *p* = 0.21) were significantly associated with improvement in BP at remeasurement (data not shown). In a multivariable model, higher genetic risk, being older than 60 years, female gender, and a higher baseline nutritional score were negatively associated with improvements in BP, albeit non-significantly (Table 3). Conversely, higher stress management and physical activity scores, as well as improved weight, were positively, but also non-significantly, associated with BP improvements (Table 3).

## 4. Discussion

In this study, we assessed the prevalence of elevated and high blood pressure in a Dutch general population cohort undergoing a preventive health screening and digitally-enabled lifestyle intervention. Prevalence of stage 1 and stage 2 hypertension was roughly 35% and 25%, respectively. Additionally, there was no significant difference in BP in participants at higher genetic risk. In the subgroup of participants with high blood pressure at baseline who participated in the digital lifestyle intervention focused on blood pressure regulation, both SBP and DBP improved significantly. These preliminary findings suggest that a digitally-enabled lifestyle intervention leads to improvements in blood pressure regulation.

A high BP, and the subsequent development of hypertension, are the result of the interaction between environmental and genetic factors. Previous research has indicated that lifestyle risk factors such as obesity, excessive alcohol consumption, sedentary lifestyle, unhealthy diet and stress contribute to the risk of hypertension more significantly than and genetic predisposition alone [5]. It is therefore natural for the incidence of poor BP regulation and hypertension to increase as the prevalence of unhealthy lifestyle habits and lifestyle-related risk factors also increases [27]. We verified this in our population, with a significantly higher prevalence of high BP than previous older reports from Dutch cohorts, which indicated approximately 14% of a Dutch cohort aged 30 to 59 years had stage 2 hypertension, but this is only slightly higher than recent data suggesting a hypertension prevalence for the 40–60 demographic to be 30.3% [28,29]. We also assessed whether genetic risk determined by polygenic risk scoring translated to higher measured BP using a validated genetic analysis pipeline. We found no difference in measured BP between participants in different strata of genetic risk. This may be due to the small cohort size, which makes it challenging to identify significant differences in genetic risk. Large-scale studies, including our own stratification model validation study, have shown evidence that individuals with a higher polygenic risk score do indeed develop hypertension more frequently over the 8 to 10 years after the initial assessment [22]. However, other studies have suggested that higher genetic risk—especially in younger populations—does not necessarily translate to higher BP at the time of the baseline assessment [30]. Similarly, we also did not find an association between baseline genetic risk and BP reduction after intervention. This indicates that the effectiveness of this lifestyle intervention is independent of baseline genetic risk, which is in line with previous studies reporting on the effects of adhering to healthy lifestyle principles similar to our intervention on BP regulation [6]. The absence of a significant effect on BP improvement of variables such as baseline dietary, physical activity, or stress management scores are more surprising, but may be explained by the small sample size of the of the study. Future studies could also explore the metrics of engagement with the coaching intervention and with the digital application as explanatory factors for responding or not to the intervention.

Evidence on the beneficial effects of healthful lifestyle modifications on BP regulation—supported by digital therapeutics or not—is overwhelming, with several studies suggesting this should be preferred to pharmacological treatment in early stages of the disease [31,32,33]. Yet, a physicians’ task of motivating patients at risk to adopt these and other lifestyle practices remains a difficult one for primary and secondary care practitioners alike. This is partly due to limited time and contact with their patients, as well as the lack of structural support needed to effectuate behavioral change [34]. Digitally-enabled lifestyle interventions have the potential to close this gap, advising and guiding patients along a behavioral change journey, without increasing the burden on care providers [35]. While literature backing the real-world effectiveness of said interventions remains limited, recent studies have suggested their superiority to standard lifestyle modification alone in improving BP regulation. In the recent landmark HERB-DH1 trial, patients assigned to a digital therapeutics group enabling lifestyle modification showed a decrease in ambulatory BP of between 2.4 and 4.3 mmHg compared to the control group receiving standard lifestyle modification alone [36]. However, physicians were allowed to also (re)start antihypertensive medication during the digital intervention, and the authors reported no differences in BP values between intervention and control groups in participants who did not also start medication. Albeit observational, our data show that even greater improvements in BP can be achieved with digital lifestyle therapeutics alone. This may be due to this intervention’s roots in a behavioral science framework, which a recent review suggests most interventions used in both study and real-world settings lack [21].

This study presents several limitations, and the preliminary results presented here must be interpreted in light of these limitations. The first is the small sample size of the remeasured population, associated with the high attrition rate registered for the intervention. Of the 209 individuals stratified to the blood pressure regulation intervention, only 56 (27%) opted to get remeasured within the reported study period. Attrition rates in mHealth studies and real-world applications have been reported to be as high as 80%, with participants either only minimally making use of the intervention, or entirely dropping out of the intervention after the start [37,38]. Based on these preliminary findings, several steps are being taken to improve the digital engagement strategies deployed in this version of the intervention in order to reduce attrition. Secondly, there may be some selection bias in this remeasured population, as remeasurement was optional and voluntary. Participants who came for a remeasurement could represent a more engaged sub-population, or represent a group who actively worked on behavioral change and therefore expected results. Lastly, this study lacked a control group, which prevents direct comparison of the effectiveness of the intervention with standard care.

Conversely, one of the strengths of the study is that medication information was gathered at baseline and follow-up. This allows us to verify that participants with improved BP did not initiate medications. In addition, despite lacking a control group, we analysed available reports of the effectiveness of combined lifestyle interventions in Dutch populations to establish a comparison between the results achieved in our intervention and a comparable intervention [39]. There, registered improvement in an intervention group of around 100 individuals was 3.5 mmHg in SBP, and 3.4 mmHg in DBP. However, comparable changes were also verified in the control group of 200 participants (3 mmHg and 3.6 mmHg, respectively, for SBP and DBP). In both cases, these improvements were inferior to the improvements we registered, which reinforces the potential of the intervention deployed in this study to yield better health outcomes than other currently available options. Finally, few studies have reported on real-world applications of digital interventions targeting health behavior change and its effects on blood pressure. With this study, we contribute to the for now scarce body of evidence for the usefulness of digital, lifestyle programs for cardiovascular risk reduction through blood pressure regulation.

## 5. Conclusions

In conclusion, we have identified a significant portion of individuals with above-optimal BP or stage 1 hypertension in a general population cohort who could benefit from lifestyle intervention for BP management. Participants who subsequently participated in a digitally-enabled, lifestyle intervention and returned for remeasurement showed significant improvements in BP compared to baseline, and without need for concomitant pharmacological therapy. Improvement was independent of genetic risk. The long-term sustainability and the efficacy of this and similar digital lifestyle interventions will need to be established in additional studies at larger scale.

## Figures and Tables

**Figure 1 ijerph-19-04171-f001:**
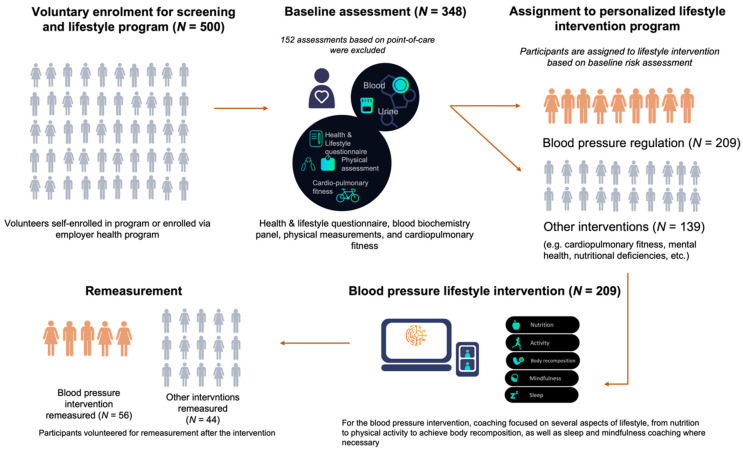
Overview of the study flow, including sample size at each stage. Changes in blood pressure presented in the results are derived from the blood pressure regulation intervention remeasurement group (*N* = 56).

**Figure 2 ijerph-19-04171-f002:**
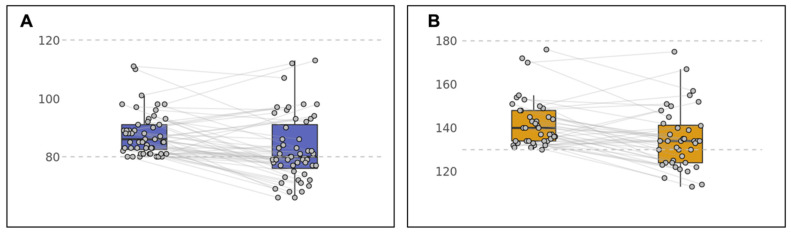
(**A**) Change in diastolic blood pressure in the intervention group with coaching focused on BP regulation; (**B**) Change in systolic blood pressure in the intervention group with coaching focused on BP regulation.

**Table 1 ijerph-19-04171-t001:** Baseline characteristics of the total study sample. Values are means with standard deviations for continuous variables, and in percentages for categorical variables.

	Entire Population (*n* = 348)	Blood Pressure Intervention Group w/Follow-Up ^a^ (*n* = 56)	*p*-Value
Demographics			
Age (years)	44.6 (11.1)	46.4 (10)	<0.0001
Sex, female	195 (56%)	22 (39.3%)	0.02
Blood pressure			
Systolic (mmHg)	131 (16.4)	137 (12.9)	<0.0001
Diastolic (mmHg)	81 (11.2)	87.4 (9.3)	<0.0001
Previously diagnosed with hypertension	13 (3.7%)	0 (0%)	0.268
Taking antihypertensive medication	6 (1.7%)	0 (0%)	0.268
Genetic risk			
High	20 (5.7%)	5 (8.9%)	0.706
Elevated	58 (16.7%)	9 (16.1%)	0.987
Not elevated	251 (72.2%)	29 (51.8%)	0.239
Not available	19 (5.4%)	13 (23.2%)	
Anthropometrics			
Weight (kg)	77.2 (14.4)	78.3 (13)	0.017
BMI (kg/m^2^)	25.0 (4.7)	24.4 (3.9)	0.003
Body fat percentage (%)	24.9 (9.8)	23.0 (9.6)	0.13

^a^ Participants with high blood pressure at baseline who underwent a lifestyle intervention with focus on blood pressure regulation.

**Table 2 ijerph-19-04171-t002:** Change in systolic and diastolic blood pressure in the intervention group.

Blood Pressure	Blood Pressure Intervention Group w/Follow-Up			*p*-Value
	*Baseline*	*After Intervention*	*Change*	
Systolic blood pressure (mmHg)	142.3 (11.3)	135.1 (13.8)	−7.2	<0.01
Diastolic blood pressure (mmHg)	88.4 (8.7)	83 (11.4)	−5.4	<0.008

**Table 3 ijerph-19-04171-t003:** Associations between different demographic, genetic, and lifestyle factors and improvement in BP after intervention (per unit increase unless otherwise stated).

Variable	β	Odds Ratio (95% CI)	*p*-Value
Sex (female)	−0.35	0.71 (0.38–1.33)	0.28
Age (above 60 years old)	−0.34	0.71 (0.09–5.4)	0.74
Stress management score	0.41	1.5 (0.63–3.57)	0.36
Physical activity score	0.31	1.36 (0.65–2.83)	0.41
Nutrition score	−0.4	0.67 (0.35–1.3)	0.24
Weight loss	0.02	1.02 (0.55–1.89)	0.95
High genetic risk	−0.15	0.86 (0.25–2.93)	0.81

β: effect estimate.

## Data Availability

The data that support the findings of this study are available from the authors upon reasonable request.

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
