# Peer review of "Effect of a Digitally-Enabled, Preventive Health Program on Blood Pressure in an Adult, Dutch General Population Cohort: An Observational Pilot Study"

_ijerph, 2022, doi:10.3390/ijerph19074171_

Round 1

Reviewer 1 Report

It is a well-written paper on risk scoring, and the effects of a digitally enabled health program on lifestyle change to influence high blood pressure.

The study is clearly defined as an observational (pilot) study, which the title might reflect.

The limitations of this study are substantial but clearly defined in its chapter. First, the high quote of 73% from the intervention program might lead to an increased selection bias, limiting the prevalence of elevated and high blood pressure in the normal and the risk groups.

Nevertheless, it describes the positive effect of the intervention, although in a selected population. The described methodology might serve as a help for other studies like this. 

In the conclusions in line 285, I suggest adding: THose, who .....lifestyle intervention and were available for measuring...

Author Response

Thank you for your comments. Please see the attachment for the point by point response.

Reviewer 2 Report

The paper presents the role of a computer application in the arterial hypertension prevention program. Due to the relatively small number of respondents, the work should be treated as preliminary research. The results presented in this study are promising.

My comments for the work are as follows :

1. It is worth specifying the tools with which the training was conducted, were they smartphones, computer?

2. An incorrect reference to the graph was given for the results of changes in blood pressure (this is Figure 1, instead of Figure 2, page 5, line183). 

3. When describing the influence of genetic risk and weight changes on blood pressure in the intervention group (page 5, lines 191-193), no reference was made to the table or graph where these data would be given. If the authors do not want to create an additional table, it is worth considering specifying in parentheses information that the data is not shown.

4. Weight reduction leads to a reduction in blood pressure, therefore the advantage of this study is that the authors assessed this relationship, although no significant relationship was found in this case (p = 0.09). On the other hand, could authors indicate if gender can also be a determinant influencing the tested parameter (BP). Gender may be a factor that influence on  motivation to do an activity (e.g., exercise). Did this study show any changes in blood pressure values in the intervention group due to this factor? 

5. It is also necessary to adjust the method of citing literature to the journal's requirements; providing the name of the first author and the year of publication of the article.

Reviewer 3 Report

The pilot study aimed to observe the effect on BP of a subsequent digitally-enabled lifestyle program. One of the strengths of the study is that the authors considered the effects of genetic factors. However, except for the limitations that the author mentions, the study remains several uncertainties for knowledge. I raise some questions as followed.

  1. Participants followed a 16-week program but were remeasured after 10 months. I wonder if the participant’s lifestyle improvement or medication condition can be continued without changs for such a long period after the intervention.
  2. Because of a small sample size the study used, a non-parameter statistics analysis may be more appropriate than the traditional pair t-test. Moreover, as being a comparison between two dependent samples, the McNemar test is suggested instead of a Chi-square test.
  3. Stage 1 and stage 2 hypertension needed to be defined.
  4. In Table 1, none participant was previously diagnosed with hypertension or taking antihypertension medication. It seems to be able to infer that the effects of the program only exist in healthy people. If so, the findings will contribute less in knowlege. 
  5. The study compares the difference in continuous BP between before and after the intervention. To give a clinical meaning, I suggest the authors compare the difference of dichotomous data (i.e. normal vs. abnormal) also.
  6. As being a non-communicable disease, hypertension may be caused by daily activity, emotion, weather, diet, lifestyle, etc. The pilot study cannot rule out the factors. A multiple-variated analysis considering these confounders will be more convincing to readers than a single variable analysis.
  7. The format of citations needs to be made sure.

Author Response

Thank you for your feedback. Please see the attachment

Round 2

Reviewer 2 Report

The authors took into account the comments and the publication may now be published.

Reviewer 3 Report

The authors have answered my questions and revised the article with effort.